# Molecular Pathways for Polymer Degradation during Conventional Processing, Additive Manufacturing, and Mechanical Recycling

**DOI:** 10.3390/molecules28052344

**Published:** 2023-03-03

**Authors:** Daniel V. A. Ceretti, Mariya Edeleva, Ludwig Cardon, Dagmar R. D’hooge

**Affiliations:** 1Centre for Polymer and Material Technologies, Department of Materials, Textile and Chemical Engineering, Ghent University, Technologiepark, 130, 9052 Ghent, Belgium; 2Laboratory for Chemical Technology, Department of Materials, Textile and Chemical Engineering, Ghent University, Technologiepark, 125, 9052 Ghent, Belgium; 3Centre for Textile Science and Engineering, Department of Materials, Textile and Chemical Engineering, Ghent University, Technologiepark, 70A, 9052 Ghent, Belgium

**Keywords:** thermal degradation, thermal-oxidative degradation, thermo-mechanical degradation, hydrolysis, degradation during processing, extrusion-based additive manufacturing

## Abstract

The assessment of the extent of degradation of polymer molecules during processing via conventional (e.g., extrusion and injection molding) and emerging (e.g., additive manufacturing; AM) techniques is important for both the final polymer material performance with respect to technical specifications and the material circularity. In this contribution, the most relevant (thermal, thermo-mechanical, thermal-oxidative, hydrolysis) degradation mechanisms of polymer materials during processing are discussed, addressing conventional extrusion-based manufacturing, including mechanical recycling, and AM. An overview is given of the most important experimental characterization techniques, and it is explained how these can be connected with modeling tools. Case studies are incorporated, dealing with polyesters, styrene-based materials, and polyolefins, as well as the typical AM polymers. Guidelines are formulated in view of a better molecular scale driven degradation control.

## 1. Introduction

During their first life cycle, polymer goods are exposed to several physical, chemical, and/or biological factors that may cause their degradation, as shown in the top right part of Figure 1. Among these factors, heat, moisture, UV-radiation, mechanical stress, chemicals, oxygen (O_2_), and micro-organisms are the most common ones [1,2,3,4,5]. At the end-of-life processing of polymer goods, the generated post-consumer waste may have different destinations, including chemical or mechanical recycling, to again obtain raw materials, landfill, or energy recovery [6]. In case the materials are reprocessed in the context of a mechanical recycling process, e.g., reshaping via conversional extrusion, they are exposed to some of the same factors (smallest green arrow in Figure 1), which may induce a more severe degradation [6]. One can also apply chemical recycling, in which the strongest degradation is desired, and one aims at monomer (or oligomer) recovery (largest green arrow in Figure 1). 

In general, multiple-step processing techniques are more prone to induce polymer degradation, a more recent example being additive manufacturing (AM) or 3D printing via fused filament fabrication (FFF). This AM technique first requires the filament fabrication via conventional extrusion and then the actual 3D printing, leading to a higher level of polymer degradation than single-step 3D printing [7,8,9]. AM is defined by ISO/ASTM as a process of joining materials to make parts from 3D model data, usually layer by layer [10]. The main advantages of AM include the near-zero waste related to processing, the possibility to produce very complex tailored parts for specific applications, the minimum need for post-processing operations, and the possibility of creating products in low amounts in an economically viable way [11]. Note that not only polymers but also ceramics, metals, and composites are suited for AM. 

Polymer degradation mechanisms are ideally classified according to the factor or combination of factors that induce it, as summarized in Table 1. Certain factors are common to processing, such as heat, mechanical stresses, oxygen, and moisture, leading to specific types of degradation mechanisms, which are part of the scope of this review. In the present work, emphasis is on thermal, thermo-mechanical and thermal-oxidative degradation, and hydrolysis. Other mechanisms commonly related to the use and disposal of polymer goods (e.g., biodegradation) are also listed in Table 1 for completeness. 

If a polymer undergoes degradation, usually irreversible structural changes take place at the molecular scale. Hence, a good knowledge of polymer chemistry is of added value to define degradation mechanisms. As exemplified in Figure 2, the degradation-induced changes are related to molecular variations such as chain length, dispersity (IUPAC recommended term, previously known as polydispersity index), presence of certain functional groups, degree of branching, and crosslinking. These changes happening on a microscopic level are responsible for variations of the polymeric material properties such as mechanical [12,13], thermal [13], and optical properties [13,14]. One can also aim at molecular improvement or chain repair by tailoring compounds which protect polymer chains against degradation [15]. These compounds, known as stabilizers, are usually incorporated into the material during processing as additives to reduce the degradation rate [16]. A broad range of stabilizers are available on the market and the needed stabilizer type depends on the type of environment against which the polymer must be protected, as well as the function of the final polymeric material [17]. Hence, the properties of polymer goods have been improving constantly, through the design of new additives and formulations [18,19,20,21]. 

Extra research efforts are still needed, as the current challenges of our society move us towards a circular economy model (CEM). According to the CEM, solid wastes should not only be minimized but also efficiently and effectively recycled to keep their value in the economy as long as possible [22,23]. In order to do so, a deeper understanding of polymer degradation mechanisms is needed to produce materials with higher levels of stability. Specifically, for the case of mechanical recycling, understanding the degradation during processing is crucial, including both conventional processing and the newer AM techniques. One of the most interesting techniques is extrusion-based AM (EAM), in which the material is selectively dispensed through a nozzle or orifice [10]. In EAM, the material is heated up above its melt temperature (*T_m_*) for semi-crystalline polymers or above its glass transition temperature (*T_g_*) for amorphous polymers. The material melts or softens, and it is pushed through the nozzle onto a build platform. A sliced 3D model is used to give the commands to the printer, which will create the part in a layer-by-layer fashion [24,25,26,27]. The feedstock or bulk raw material for EAM may be filaments, pellets, or solutions depending on the nature of the AM. FFF is the most common EAM technique, followed by pellet-based additive manufacturing (PBAM) and syringe-based additive manufacturing of which the main principles are highlighted in Figure 3.

As EAM resembles conventional processing via melt extrusion, it is mostly used to process thermoplastics and composites to, for instance, produce prototypes and functional components for medical applications, or components for construction, aerospace, architecture, and automotive industries [28,29,30,31,32]. Furthermore, formulations of highly-filled polymers with metals or ceramics have been printed [33], as have polymer solutions and hydrogels using a syringe-based 3D printer [34,35]. 

The aim of the current review is to summarize the advances in the study of polymer degradation during conventional processing, including mechanical recycling, and EAM. We initially focus on the common degradation pathways involved in polymer processing. Further, we discuss the degradation mechanisms of selected polymers that are widely used under EAM conditions and shared with conventional processing. This first part is followed by a second part, summarizing the most employed experimental and modeling techniques to characterize polymer degradation. In the third part, case studies of degradation during processing are considered for polyester, styrene-based, and polyolefin materials, both under conventional and AM processing conditions.

## 2. Overview of Common Degradation Reactions

Processing techniques, such as extrusion, injection molding, and EAM, usually deal with relatively high temperatures and shear rates, with a certain (limited) amount of oxygen solubilized in the polymer [36]. Temperature, mechanical stress, and oxygen are the relevant factors inducing polymer degradation during polymer processing and mechanical recycling, promoting thermal, thermo-mechanical, and thermal-oxidative degradation, as described in Table 1. Furthermore, hydrolysis may take place, due to the combination of moisture and temperature, especially for polymers prepared via step-growth polymerization or polycondensation [37]. 

Thermal (radically induced) degradation occurs if the polymer bonds break due to heat, either supplied externally or generated by shearing via heat dissipation [38,39,40,41,42]. Table 2 lists representative bond dissociation energies (BDEs) between common atomic constituents of a polymer. The polymer thermal stability increases if the BDEs are higher. The BDE is not an exclusive discriminator as the reactivity of the de-propagating radicals and the availability of hydrogen atoms for chain transfer also matter [43,44,45,46]. 

The major pathways for thermal degradation are chain fission (formally scission; no radicals at the reactant side, consistent with, e.g., the work of the Broadbelt group [47]), end-chain β-scission (so-called depolymerization), side-group elimination, and hydrogen abstraction or chain transfer [15,38]. Thermal chain fission is an event in which the polymer chain is randomly broken into large macroradicals (Figure 4a), leading to a fast decrease in average molar mass (also often referred to as molecular weight despite not being IUPAC recommended), with almost no monomer formed in the early stages [15,48,49,50,51]. End-chain β-scission takes place at chain ends to split off monomers sequentially (Figure 4b), leading to a very small decrease in the average molar mass in the early degradation stages [15,48,49,50,51]. The contribution of chain fission versus end-chain β-scission depends on the polymer structure and is closely related to the amount of substituents on the alpha carbon, with a more stabilized case leading to more end-chain β-scission [15,48,49,50,51]. Side-group elimination refers to elimination of side groups attached to a backbone of the polymer [50,52]. This process can result in main chain unsaturation, crosslinking, and volatile product formation [53,54]. In addition, the formation of aromatic molecules, fission into smaller fragments, or the formation of char is also possible [38].

As shown in Figure 4c, hydrogen abstraction generates a mid-chain radical (MCR), which is prone to further degradation reactions. Mechanical stresses may additionally favor degradation by breaking the chains and generating radicals via a mechanochemical mechanism [55] (Figure 4d), lowering the average molar mass. The formed radical will be available for other reactions as illustrated in the left part of Figure 5, for illustration purposes starting from an MCR. The right part of this figure also puts forward the extra reactions in case oxygen is present, defining thermal-oxidative degradation, starting from the same MCR. In general, the radicals are formed due to heat, trace impurities, catalysts, inhibitors, solvents, and other agents used for the synthesis [56]. Notably, during processing, thermal-oxidative and thermo-mechanical degradation may be stimulated by one another. For instance, radicals produced via thermo-mechanical degradation may readily react with oxygen and increase the initiation rate of the thermal oxidation pathway, as highlighted by the dashed arrows in Figure 5 [17,57].

As shown in the right part of Figure 5, polymer radicals formed due to thermal degradation react with molecular oxygen, delivering oxidation products based on an autoxidation cycle [59]. The thermal oxidation mechanism proceeds autocatalytically, as shown by Bolland and Gee [60,61,62] upon studying the oxidation of rubber and lipids, thus unsaturated chains. These authors indicated that the radicals propagate with oxygen to form a peroxyl radical (ROO^•^), which in turn abstracts a hydrogen from another macromolecule. This transfer leads to a new chain, in which the newly formed radical center is stabilized by the conjugation with the adjacent double bonds, as shown in Figure 6 [63]. 

In many cases, the reactions in Figure 6 have also been seen as likely for several polymers with a saturated carbon chain backbone. However, Gryn’ova et al. [63] put forward that the hydrogen abstraction by the peroxy radical should be (except for unsaturated hydrocarbons) thermodynamically disfavored, as the formed product is less stable than the peroxyl radical. They suggested that structural defects in polymers are responsible for the propagation in the autoxidative cycle, mostly starting from terminal and internal double bonds, which may be formed during the original polymerization procedure and/or the degradation process itself [63]. Hydroperoxide species are assumed as key intermediates in Figure 5 to further reactions, as they quickly decompose to form alkoxy (RO^•^) and hydroxy (^•^OH) radicals, leading to the backbone degradation followed by β-scission, as shown in Figure 7 (complementary to reactions of Figure 5) [64].

The termination step for Figure 5 depends on the type of atmosphere in which the degradation is occurring. If oxygen is sufficiently available, recombination of peroxy radicals has been put forward as the most probable termination mechanism [59]. During processing, due to the deficient oxygen atmosphere, the concentration of alkyl radicals is higher than the peroxy radicals, and the possibilities for termination are likely conventional termination by recombination and disproportionation of C-centered radicals, as shown in Figure 8.

As a result of thermal-oxidative degradation during polymer processing, new oxygenated functional groups are formed, including peroxides, alcohols, ketones, aldehydes, acids, and esters [59]. Ketones and aldehydes formed by β-scission may react further with hydroperoxides to form other oxidation products. For instance, the formation of peracids proceeds via a multi-step oxidation of aldehydes [65]. Oxygenated functional groups can be formed even in carbon main chain polymers, such as polyolefins [66,67] or styrene-based materials [68]. 

Finally, Figure 4e highlights the mechanism of hydrolysis, which can occur if moisture is present in a high temperature environment. Certain polymers, especially those containing heteroatoms in the backbone are prone to this degradation mechanism and must be properly dried before processing [37]. Polyesters, polyamides, and polycarbonates are among the important commercial polymers that could be degraded by hydrolysis [15]. This degradation pathway leads to the cleavage of chemical bonds by reaction with water, reducing the average molar mass of the polymer [69]. 

## 3. Dominant Degradation Reactions Illustrated for Extrusion-Based Additive Manufacturing

The most used polymeric materials for EAM are acrylonitrile butadiene styrene polymer (ABS) [70,71] and poly(lactic acid) (PLA) [70,72,73]. Other polymeric materials which have been employed for EAM applications are polyethylene terephthalate glycol (PETG) [70,72,74,75], polyether ether ketone (PEEK) [70,76,77,78,79,80,81], high density polyethylene (HDPE) [82], polypropylene [83,84,85], polyamide (PA) [86,87], and high impact polystyrene (HIPS). The main chemical structure of these polymeric materials and their most relevant degradation routes during EAM, starting from the discussion in Section 2, are summarized in Table 3. 

ABS (entry 1 in Table 3) polymer consists of a matrix based on styrene-acrylonitrile (SAN) copolymer with polybutadiene (PB) rubber dispersed in it, supported by SAN-grafted-polybutadiene (PB-g-SAN) connections. The ABS degradation mechanism is closely related to the degradation paths of its individual components [100]. However, the rubber phase is more susceptible to degradation because of its lower glass transition temperature *T_g_* and higher oxygen permeability [101,102,103]. 

During (additive) manufacturing, the most common degradation route for ABS is proposed to be thermal oxidation [104,105,106]. An interesting off-line study has been performed by Shimada and Kabuki [107], mentioning the three stages of initiation, propagation, and termination, as included in the auto-oxidative cycle in Figure 5. The first stage involves the production of free radicals which react with oxygen (Figure 9a) and form hydroperoxides via hydrogen abstraction (Figure 9b). Because of the presence of ethylene functionalities in the PB phase of ABS, the C-H bond in the α-carbon position can be easily oxidized [107]. The hydroperoxides decompose and in following reactions various oxygenated products and groups, particularly hydroxyl and carbonyl, are produced (Figure 9c). In the third stage, termination reactions occur, in which more stable crosslinked structures may be formed, with one example shown in (Figure 9d) [108]. The butadiene grafting sites containing tertiary carbons can be oxidized during a more severe degradation, leading to extensive damage in the polymeric chains by β-scission, as well as breaking of butadiene/SAN grafts [109,110]. 

PLA (entry 2 in Table 3) is a linear, aliphatic thermoplastic polyester with high rigidity and transparency. PLA is also a biodegradable polymer but with low thermal stability. Due to its hygroscopic nature, hydrolysis is one of the main degradation mechanisms for this polymer (see Figure 10a) [111]. Therefore, the material must be properly dried prior to processing [112], which is a common procedure in many EAM studies [113,114,115]. In addition, McNeill and Leiper [116] discussed that radical reactions for PLA undergoing thermal degradation only occur above 270 °C. They proposed that non-radical degradation mechanisms occur up to 230 °C, as shown in Figure 10b,c. These reactions and hydrolysis end up reducing the average molar mass of the PLA material. Nevertheless, radical reactions could occur, even below this threshold temperature of 270 °C, due to the combination of shear and oxygen [117,118]. Figure 10d, for instance, shows the reaction mechanism for the development of hydroperoxides during the thermal-oxidative degradation of PLA. 

Additional interesting polymer materials to discuss are polyolefins (entry 3 in Table 3), comprising materials such as polypropylene (PP), high-density polyethylene (HDPE), and low-density polyethylene (LDPE). These materials are composed of carbon and hydrogen atoms and during degradation chain fission, chain branching (e.g., via macropropagation), and crosslinking are possible. It has been claimed that more branched structures such as LDPE are more susceptible to crosslinking or chain branching reactions during extrusion compared to more linear polymers [95,119]. 

The degradation mechanisms of PE, in the presence of a low amount of oxygen has been studied by Holmström and Sörvik [120,121,122], and typical reactions are summarized in Figure 11. Initiation degradation reactions are shown in Figure 11a. The quick formation of peroxide radicals is followed by hydrogen abstraction with hydroperoxide formation (Figure 11b). Subsequent reactions (Figure 11c) result in the formation of aldehydes, ketones, or hydroxyl compounds and other alkyl radicals. The reaction of a polymer chain with formed radicals leads to an auto-oxidation (Figure 11d), for which R• may be an alkyl radical, RO•, ROO•, or HO•. Important termination reactions involving alkoxy radicals are shown in Figure 11e. Other termination pathways leading to an increase of the average molar mass, including chain branching, are reported in Figure 11f. 

Due to the presence of tertiary carbons in the backbone of PP (see Table 3), in contrast with PE, its degradation mechanism is chain fission driven [123,124], decreasing the average molar mass. Figure 12 gives an overview of typical reactions for PP degradation, again in the presence of oxygen. Figure 12a displays a degradation initiation with the radical formed reacting with oxygen in Figure 12b. Hydroperoxide is formed by abstracting a hydrogen from a tertiary carbon, which has a lower activation energy than an abstraction reaction on a secondary carbon [65]. This preferred abstraction can be intermolecular (Figure 12c) or intramolecular (Figure 12d). Furthermore, the radicals may decompose via β-scission, yielding, for instance, ketones, aldehydes, and other alkyl radicals (Figure 12e), with termination reactions for low-oxygen-containing atmospheres shown in Figure 12f. 

## 4. Experimental and Modeling Techniques to Assess and Quantify Polymer Degradation

Irreversible structural changes of polymers are typically investigated by a set of experimental characterization techniques which provide information on the type of degradation mechanism by tracking the molecular properties (e.g., chemical composition, average chain length, chain length distribution, and branching and crosslinking level), the material properties (e.g., thermal, tensile, and flexural) and the morphology variations (e.g., single vs. multiphase). These experimental characterization techniques include chromatography, rheometry, viscometry, spectroscopy, thermal analysis, microscopy, and mechanical testing, with a summary provided in Figure 13a. 

Complementary (kinetic) modeling can be applied in which typically average properties are compared to experimental data [125,126,127,128,129,130,131,132,133,134,135,136,137] and subsequently the model is employed to provide additional information that is hard to access experimentally. A differentiation can be made between deterministic and stochastic modeling tools. With increasing computer capacity, specifically in the recent decades, Monte Carlo methods have become specifically interesting. For example, as shown in Figure 13b, coupled matrix-based Monte Carlo simulations [125,126] allow one to track the functional groups, branches, and monomer sequences of individual chains as they come out of a processing unit, allowing a more detailed understanding of the specific molecules that are being degraded, e.g., one can determine if the attack is happening in the lower or higher chain length region and if the mixing pattern in the processing equipment is altering this division. Moreover, a broad range of processing conditions can be scanned a priori to assess the degradation potential of a given polymeric material either under conventional or AM processing conditions, including also conditions leaning toward full degradation, and thus, chemical recycling.

In what follows, a concise description is given of the three experimental characterization groups in Figure 13a, making a link to available theoretical tools.

### 4.1. Molecular Properties Characterization 

Chromatography, rheometry, and viscometry are the most important characterization techniques employed in the assessment of degradation during processing, as they are very sensitive to (average) molar mass variations [139]. Size exclusion chromatography (SEC) provides the dispersity (*Đ*), the average molar masses of the polymer such as the number average molar mass (*M_n_*), the mass average molar mass (*M_m_*), and the *z*-average molar mass (*M_z_*), as well as the molar mass distribution. Nevertheless, static light scattering and osmometry may also be used to determine *M_m_* and *M_n_*, respectively [140]. These techniques are especially useful in case the polymer has an insoluble fraction in its structure, caused by crosslinking or branching reactions during degradation, leading to the formation of a microgel.

Furthermore, linear viscoelastic measurements in a parallel plate rheometer or capillary rheometer allow to obtain flow curves, which via extrapolation allow the determination of the zero-shear melt viscosity (*η*_0_). This viscosity parameter may be correlated with *M_m_*, for linear polymers often expressed by an empirical power law ($\eta_{0}=KM_{m}^{\alpha }$; *K*: proportionality factor; *a* = 3.5 ± 0.2) [141]. A similar expression named the Mark–Houwink(–Sakurada) (MH(S)) equation is used to estimate the average molar mass from intrinsic viscosity measurements in solution ($\left[\eta \right]=K_{m}M_{m}^{a})$ [141]. In general, more complex relations can be obtained via rheokinetic analysis, which associates the reaction kinetics (e.g., degradation) with rheological constitutive models [142,143,144].

Another parameter of importance for degradation analysis obtained via rheometry is the relaxation time (λ), which is related to the molecular structure of the material [145]. The dispersity of polymers may also be assessed via rheological analysis either via the storage modulus (*G*’) or via correlations with viscosity [141]. This may be especially useful to estimate the dispersity of polymers which are difficult to analyze via SEC [146,147]. Another technique commonly applied to give a qualitative insight on the degradation of the materials is the melt flow index (MFI). The MFI value indicates the easiness that the material flows. If there is a decrease in the average molar mass of the material, the tendency is that it will flow easily, increasing the MFI. 

Spectroscopic techniques measure the behavior of a material in response to electromagnetic irradiation. The most common tool is Fourier-transform infrared (FTIR) spectroscopy, which is a vibrational spectroscopic technique. The different types of interatomic bond vibrations existing in organic molecules, polymers, and composites have been investigated [148] so that it is possible to identify and track changes regarding functional groups present in the polymer, e.g., because of degradation. Specifically, carbonyl (>C=O) and hydroxyl (-O-H) vibrational regions are of interest upon analyzing the FTIR spectra of polymers which underwent degradation, located around 1600–1800 cm^−1^ and 3200–3600 cm^−1^, respectively [65]. For instance, the rate of carbonyl formation can be used to follow oxidative degradation in some polymers [15,149]. In any case, a dedicated computational analysis of spectra is needed [150].

Similar to FTIR, Raman spectroscopy is classified as a vibrational spectroscopic technique, which may be used to monitor chemical species. In this case, the inelastic scattering of light is used to analyze vibrational and rotational modes of molecules [148]. Raman spectroscopy has also been employed to in-line monitor the degradation of PP during processing [151]. In-line characterization methods allow one to determine degradation quickly without destroying the material. Another interesting spectroscopic technique is nuclear magnetic resonance (NMR), which is one of the most important tools for the analysis of substances, specifically in the polymer synthesis field. NMR may resolve the structure of the polymer, and has therefore been used to probe the chemical variations during their degradation. For instance, the crosslinking degree in polymers undergoing degradation has been assessed [88,152,153]. Likewise, the detection of new functional groups formed due to degradation can be evaluated [154,155]. NMR can provide exact oxidation levels in a material but is more limited in the cases where solid samples (or very high molar mass samples) need to be analyzed, or the oxidation levels are low [156]. In addition, dynamic NMR may be used to probe specimens which undergo physical or chemical changes with time [38]. More information on NMR techniques can be found elsewhere [157,158].

It is important to point out that some (spectroscopic) techniques do not necessarily identify changes in the molecular structure of degraded samples in the early stages of degradation. For instance, FTIR and NMR are popular methods for degradation characterization; however, they can be ineffective at defining small structural changes [139]. Furthermore, sometimes the oxygen content inside the extruder is low enough to avoid the formation of measurable carbonyl quantities by the FTIR technique [159]. Therefore, upon studying low extents of degradation as in the context of mechanical recycling, rheometry and chromatography may be the most sensitive tools in identifying the molecular changes. In any case, the complementary information obtained by modeling tools is highly relevant.

### 4.2. Morphology Characterization

Variations on the morphology of polymers because of degradation can be detected with microscopic techniques such as optical microscopy (OM), scanning electron microscopy (SEM), and atomic force microscopy (AFM) [13,160]. Furthermore, X-ray diffraction techniques may be used to resolve morphological features such as size, distribution, and orientation of crystallites and lamellae of polymers, contributing to the study of their degradation [161]. The modeling field is less developed here compared to the molecular/micro-scale, which is understandable as in general our knowledge regarding chemical/material processes is still more basic on the so-called meso-scale. Interesting modeling approaches have although been developed; for instance, the work of Vonka and Kosek [162] puts forward that the morphology evolution of hetero-phase polymers that undergo phase separation and inversion during their formation can be captured. These authors studied the phase separation occurring in high impact polystyrene (HIPS) during mixing, indicating phase inversion settling as a function of the frequency of mixing. 

### 4.3. Material Properties Characterization

Thermal analysis is broadly applied for the study of polymer degradation. For instance, thermogravimetric analysis (TGA) allows tracking variations on the thermal stability of polymers and the onset of degradation either in the absence or in the presence of oxygen [15,163,164]. It is also possible to study the degradation kinetics of polymers subjected to different heating programs with this technique [165]. 

Furthermore, differential scanning calorimetry (DSC) is commonly used to monitor variations of *T_g_*, *T_m_*, and crystallization temperature (*T_c_*). In addition, for semi-crystalline polymers, the enthalpy of fusion (Δ*H_m_*) is used to calculate their degree of crystallinity (*χ_c_*). However, the same lack of sensitivity reported for NMR and FTIR to track early stages of degradation may be evident for thermal analysis techniques such as DSC and TGA [139].

Additionally, ultraviolet-visible (UV-Vis) spectrophotometry is a technique commonly employed to measure color variations of polymers which underwent degradation. For instance, so-called yellowing is an usual degradation outcome for manufactured polymer parts [166] and the color properties obtained via UV-Vis may be translated into a Yellowness Index (YI), which supports the assessment of degradation of the material. For instance, this technique has also been used in-line to track color changes of poly(L-lactic acid) (PLLA) undergoing extrusion [167].

Degradation has a strong influence on the mechanical properties of polymers as well. For example, predominance of crosslinking over chain fission increases the tensile strength while decreasing the elongation at break [65]. The reverse effects are observed for dominant chain fission. The influence of degradation on the mechanical properties is usually assessed via impact and tensile tests, tracking the impact resistance, tensile strength, tensile modulus, and elongation at break of the polymer [168,169,170]. 

In addition, water contact angle measurement is a valuable tool to assess polymer degradation, by providing an indication on the hydrophobicity/hydrophilicity of the polymer (surface). Higher contact angles indicate that the material has a higher hydrophobic character. Oxygenated functional groups formed due to degradation may alter this character and the material becomes more hydrophilic, exhibiting smaller contact angles [171]. Furthermore, changes in the surface morphology of the material due to degradation may also change its hydrophobic character [172]. 

## 5. Manufacturing Case Studies to Assess Degradability

The influence of single extrusion, multiple extrusion, AM, and injection molding on polymeric degradation has been broadly studied in the past decades. For instance, degradation has been assessed for PLA and PLLA [167,173,174,175,176], poly(ethylene terephthalate) (PET) [177,178], PS [42,179,180,181], ABS [182,183,184], HIPS [185,186,187], PP [93,151,159,188,189,190,191], HDPE [139,188,192], LDPE [140], and poly(methyl methacrylate) (PMMA) [42]. To highlight the advances made, an overview of the most important findings is given in this section. In case study format, polyesters, styrene-based materials, and polyolefins are considered. A special subsection is also devoted to additive manufacturing.

### 5.1. Manufacturing of Polyesters 

Taubner et al. [173] investigated the influence of moisture, processing temperature, and screw speed on the degradation of PLLA during melt extrusion. The authors identified a higher level of degradation in cases when elevated temperatures and low screw speeds were employed. This was attributed to the increased residence time of the material in the extruder and the activated nature of degradation reactions. Wang et al. [167] also showed that by increasing the residence time of PLLA in the extruder, the degradation becomes more pronounced. Furthermore, they highlighted that the decrease of the average molar mass of the material could be mainly attributed to thermal degradation for a dried material. In cases when the material is not dried, hydrolysis also plays a role for the degradation, further decreasing its average molar mass. 

Mysiukiewicz et al. [174] investigated the effects of extrusion temperature and screw speed of a co-rotating twin-screw extruder on the properties of different grades of PLA. The materials underwent thermo-mechanical and thermal-oxidative degradation under high processing temperatures and low screw speeds, again due to the increased residence time condition. Structural changes were monitored via rheological measurements, perceived by the significant decrease in the zero-shear viscosity under these conditions. Aldhafeeri et al. [175] discussed that the processing parameters which increase the residence time, i.e., lower screw rate, lower feed rates, a quad screw extruder, and kneading blocks, result in a more pronounced degradation of PLA, confirmed by the reduction in the zero-shear viscosity. Additionally, the PLA grades with a low starting viscosity were less susceptible to degradation than the ones with a high starting viscosity at higher screw speeds [174]. In addition, Oliveira et al. [176] studied the effect of temperature, shear, and oxygen on the degradation of PLA. The authors found out that thermo-mechanical degradation induced a higher degree of irreversible structural changes for the material than thermal-oxidative degradation. 

Alongside PLA, emphasis has also been on the processing of PET. For example, Spinacé and de Paoli [177] studied the effects of multiple extrusion cycles on PET properties. They observed a dramatic change of the mechanical properties after few extrusion cycles. The MFI, carboxylic end group content, and color properties changed right after the first processing cycle, while no changes in thermal properties could be recorded. Al-AbdulRazzak and Jabarin [178] in turn assessed the effects of moisture, temperature, and type of resin on the degradation of PET processed via injection molding. The authors concluded that the contribution of the distinct types of degradation to the overall (average) molar mass decrease depends on the processing conditions. For a high moisture content, the most important degradation mechanism is hydrolysis. In contrast, for low moisture contents, thermal and thermal-oxidative degradation become the most important degradation mechanisms. Additionally, material grades with a higher initial carboxyl content led to higher levels of carboxyl end-group generation because of hydrolysis. In turn, the rate of hydrolysis increased due to an autocatalytic mechanism promoted by these newly formed groups.

### 5.2. Manufacturing of Styrene-Based Materials

Whitlock and Porter [179] studied the source of degradation of PS during its extrusion in a capillary rheometer. The role of the temperature, initial average molar mass, and type of atmosphere were evaluated. At higher processing temperatures, the decrease in the average molar mass is more pronounced, indicating a more severe degradation. The measurement of the average molar mass at different concentric sections of extrudates indicated the existence of an (average) molar mass gradient throughout the specimen, with a reduced molar mass close to barrel, in comparison to the middle of the extrudate, which exemplifies the effect of shear on the degradation process. The authors also put forward that differences between the initial molar mass affect the rate of degradation, as the highest molar mass species were most subjected to shear-induced degradation. In addition, processing the materials in air led to a higher level of degradation, compared to processing under nitrogen, highlighting that air present in any conventional processing scheme may play a prominent role in the degradation of PS. 

La Mantia et al. [180] processed PS at different screw speeds, temperatures, and processing times, evidencing that by increasing the shear rate, the degradation of the material was also increased. They also corroborate the work of Whitlock and Porter [179], stating that processing in air caused a more severe degradation, due to a combined action of oxygen, temperature, and mechanical stresses. 

Capone et al. [42] explored the degradation of PS (and PMMA) during processing at different temperatures and using different screw speeds. They also evaluated the thermal stability of the polymer in the capillary rheometer, showing that in the absence of mechanical stresses and limited contact to oxygen, thermal-oxidative degradation is insignificant at temperatures below 210–230 °C. The authors pointed out that a low screw speed may induce the degradation of PS partially due to longer residence times. Finite element method (FEM) simulations suggested that at the highest rates, wall slippage phenomena may occur, which reduce the actual mechanical stress in the extruded polymers and may be a reason why the degradation level is smaller at high shear rates than first expected. In addition, the increase in temperature led to a reduction in the viscosity of the materials, confirming that combined effects of temperature and mechanical stresses can accelerate degradation in a synergistic mode. Furthermore, an estimation of the temperature along the barrel and in the channel sections via simulation showed a continuous temperature increase up to the extruder head, corresponding to output temperatures 20 to 60 °C higher than the set temperature. 

Upon assessing ultrahigh speed extrusion of PS, Farahanchi et al. [181] also measured an overshoot in the processing temperature. In their case, it was as high as 33 °C at the highest rotation speed employed (4000 rpm). Because of the high screw speeds used in their study, the mechanical stresses were the key factor causing a decrease in the viscosity and average molar mass of PS. 

Other styrenic polymers that have been investigated in terms of degradation during processing are ABS and HIPS, which are rubber-modified polymers, as described before. For instance, Karahaliou and Tarantili [182] investigated the effect of multiple extrusion cycles on ABS degradation. Even though the material exhibited yellowing and variations in FTIR spectra due to oxidation, no strong variations for its mechanical properties and MFI were witnessed. Similarly, Boldizar and Möller [183] reported minor degradation for ABS after multiple extrusion cycles, with only an increase for the elongation at break of the material. On the contrary, Salari and Ranjbar [184] reported degradation of the polybutadiene (PB-g-SAN) phase of ABS as the most important aspect to influence the properties of the material after multiple processing cycles, leading to yellowing, reduced elongation at break, lower impact strength, and an increased MFI. In addition, Kalfoglou and Chaffey [185] investigated the relevance of the processing temperature and the use of successive processing on the degradation of HIPS. During regular processing, the properties of HIPS changed only to a small extent. A decrease in elongation at break and impact resistance for HIPS after several processing cycles, or at high processing temperatures, was reported and attributed to the variations of the morphology of the material. A decrease at the interface quality between both PB and SAN phases was highlighted because of shearing, leading to agglomeration of particles, directly influencing the properties. Furthermore, the PS matrix also underwent degradation, presenting a reduced average molar mass. 

Vilaplana et al. [186] also assessed the influence of multiple extrusion cycles on the degradation of HIPS. They reported that (multiple) processing does not strongly affect the material properties, suggesting a low degree of degradation. Nevertheless, a slight increase in the MFI, with a reduction for the elongation at break and an increase in tensile strength were reported after few extrusion cycles. Furthermore, it was discussed how changes in the chemical structure of HIPS were induced because of processing, with the formation of oxidative moieties and the consumption of part of the unsaturations. Parres and Crespo [187] also indicated that the extrusion process of HIPS slightly affects its mechanical and thermal properties. The authors highlighted the design of new materials less prone to degradation, with the addition of styrene–butadiene rubber (SBR) and styrene–ethylene–butylene–styrene (SEBS).

### 5.3. Manufacturing of Polyolefins

Hinsken et al. [188] studied the thermal-oxidative and thermo-mechanical degradation of PP and HDPE during multiple extrusions. For PP, it was found that the average molar mass decreases after every extrusion pass via chain fission, resulting from the β-scission of alkoxy radicals, the breakdown of peroxy radicals, and shear. The first extrusion pass promoted the most significant decrease in the average molar mass, and this may be influenced by the increased shear (due to the high molar mass), as previously discussed for PS. 

A similar study was conducted by Da Costa et al. [189], although with a deeper focus on the rheological property changes of PP undergoing multiple extrusion cycles. The rheological properties of PP change after multiple cycles of extrusion, not only by a viscosity decrease but also by the reduction of the elasticity of the polymer melt. The authors pointed out that the factors which are important in determining the rate of degradation are the average residence time and residence time distribution. The effect of the screw element types on the degradation of PP was also assessed. Moreover, Canevarolo and Babetto [190] reported that use of kneading blocks at 90° caused a higher degree of degradation to the material compared to left-handed conveying elements. This was attributed to a lower oxidation level, because of the lower amount of oxygen available inside the barrel. 

More recently, the degradation of PP has been monitored in-line via Raman Spectroscopy [151]. The results showed a good relationship with SEC measurements, MFI, and tensile properties, indicating that free radicals were terminated with oxygen-containing functional groups instead of methyl groups, and that long chains were more easily damaged in the extrusion. 

Gonzáles-Gonzáles et al. [93] assessed the effects of extrusion temperature and extrusion cycles on the degradation of PP as well. While not evidencing oxidation during extrusion of PP, as seen via FTIR, the authors observed a big drop of the average molar masses and a lowering of the dispersity after each extrusion cycle. This effect was more pronounced at higher temperatures. da Costa et al. [191] carried out a similar study and they reported that degradation is more pronounced at high shear levels than at high temperatures, with a strong influence on the thermal properties and break properties due to the degradation of the material via chain fission. 

Upon studying HDPE, Hinsken et al. [188] showed that chain branching and crosslinking reactions resulted in an increase of the average molar mass of the polymer. These reactions were favored compared to chain fission, which was happening simultaneously, in contrast with what was observed for PP. Moss and Zweifel [192] also studied the effects of multiple extrusion in HDPE and discussed that depending on the type of HDPE used, the degradation could be either via chain fission or crosslinking, resulting in a decreased or increased average molar mass, respectively. Their study reveals that vinyl group reactions are primarily responsible for the average molar mass increase, as the increase correlates with a higher content of vinyl groups originally present. 

The increase in viscosity of HDPE due to processing has also been reported in injection molding by Rex et al. [139] and was also attributed to degradation via crosslinking, branching, or chain extension. The authors put forward several important factors for the occurrence of fission, namely a low terminal vinyl group content, the presence of oxygen in the environment, and a high average chain length. Consistently, they highlighted that branching and crosslinking are favored over fission due to the oxygen deficient conditions of the injection molding process, limiting the transformation of alkyl radicals into peroxy radicals, as well as due to the relatively short average polymer chain lengths used. 

The degradation of LDPE has also been subject of research. For example, Zatloukal et al. [140] observed that the average molar mass of LDPE gradually increased after different processing cycles. The authors discussed that chain fission (scission), branching, crosslinking, and gel formation occur simultaneously, with chain fission prevailing in processing for one or two cycles and microgel formation occurring after this. The authors indicated that the prevailing degradation mechanism of LDPE is the formation of more supramolecular structures via crosslinking, which ultimately form microgel particles. 

Hence, depending on the type of polymer being manufactured, the material and processing parameters may promote chain fission, branching, or crosslinking during processing. Overall, it can be stated that processing parameters which increase the residence time of the material will lead to degradation (cf. Figure 14), that processing temperatures must be carefully chosen, and that the materials must be properly dried to reduce degradation. In addition, the combination of these parameters may act synergistically, leading to a more severe degradation. 

### 5.4. Additive Manufacturing

Jagenteufel et al. [193] studied the degradation of ABS and PP during FFF and they concluded that the printing process resulted in negligible degradation in both materials, because of the short exposure time to high temperatures. Nevertheless, Wojtyła et al. [194] put forward that materials commonly used for 3D printing may emit both volatile organic compounds (VOCs) and ultrafine particles during processing. The authors state that ABS can be a source of potentially harmful VOCs in the temperature range of the FFF process. 

Fernandez et al. [195] assessed the influence of the filament production, the printing temperature, and the printing velocity on the degradation of copolyesters. It was observed that printing at lower velocities and at higher temperatures led to an increased degradation of one of the material grades assessed. The temperature was the most important factor in the degradation, as seen by the dramatic decrease in complex viscosity and relaxation time for this material, implying a decrease in its average molar mass. Nevertheless, the mechanical properties of the printed parts were not reduced as an effect of degradation as the coalescence between deposited layers was slightly improved at higher temperatures. However, the printing temperature parameter can only be tuned to a certain extent, as a further increase in printing temperature may result in polymer degradation, when the onset temperature for degradation is reached [26,113,196,197]. 

Other materials used in EAM often exhibit low thermal stability and are prone to degradation, limiting their processing parameters and recyclability. Spoerk et al. [198], for example, assessed the thermal stability of commercial filaments for FFF by means of rheological time sweep tests. They showed that PETG, thermoplastic copolyester elastomer (TPC), and PLA exhibit a decrease in viscosity after few seconds of testing, indicating a low thermal stability, whereas ABS exhibited an increase in the viscosity with time, probably due to the degradation of polybutadiene phase via crosslinking. Furthermore, Ahlinder et al. [199] evaluated the process induced degradation for aliphatic polyesters, PLLA, poly(ε-caprolactone) (PCL), and copolymers for tissue engineering applications, i.e., poly (ε-caprolactone-co-L-lactide) (PCLA) and poly(L-lactide-co-trimethylene carbonate) (PLATMC), that commonly undergo degradation during processing. Due to the low shear rate in the extruder (2 s^−1^), their process was mostly controlled by temperature. The degradation was monitored via SEC, evaluating the *M_n_* variation of the materials before and after each processing step. PLLA showed a reduction in *M_n_* of approximately 12 and 17% from pellet to filament and from pellet to 3D printed scaffold, respectively. The changes in *M_n_* from pellets to 3D printed scaffold utilizing PCL, PCLA, and PLATMC were 3, 6, and 4%, respectively, indicating low levels of degradation for these materials. The authors concluded that by tailoring the extrusion and FFF processing parameters, it is possible to use FFF for scaffold fabrication of the materials investigated. 

Gradwohl et al. [200] evaluated the type of EAM process on the degradation of poly(L-lactide-co-glycolide) (PLGA) copolymer. In the first process, the polymer was extruded into a filament which was further used to manufacture 3D printed parts in a commercial printer via FFF. In the second process, the pellets were directly fed into a pellet-based 3D printer (PBAM) to obtain 3D printed parts. These authors observed a decrease in *M_n_* and *M_m_* on the specimens fabricated via both processes. The *M_n_* of the pellets was reduced by 19 and 26% for FFF and PBAM, respectively. The more pronounced degradation via PBAM was attributed to the longer residence time of the material in the micro-extruder, related to the lower printing speed employed. Furthermore, successive prints exhibited even more lowered *M_n_* and *M_m_* values for the PBAM process, indicating that the PBAM settings should be further optimized. This was also observed for syringe-based 3D printing of PLGA based scaffolds, which induced more severe degradation than scaffolds produced via FFF in cases of printing with too-long melting times [201]. 

La Gala et al. [202] recently compared the properties of ABS and PLA produced via FFF and PBAM. Via FTIR, a higher degree of degradation of ABS manufactured via FFF compared to PBAM has been reported. No significant differences regarding degradation were found for PLA processed via both techniques. In a follow-up study, the effects of printing parameters and technique on the degradation and macroscopic properties have been studied for PS and ABS [7]. The FFF process generated for ABS more fission (scission), as the mechanical loading in the filament production step is high, whereas in the PBAM process more crosslinking was evident, thus a higher thermal loading took place. With PS, fission always occurred with FFF more because it uses two processing steps instead of one, specifically for higher printing temperatures. 

## 6. Conclusions

We have reviewed the most important degradation pathways for polymers during processing, including the virgin production, the mechanical recycling, and the additive manufacturing, mainly differentiating between thermal, thermo-mechanical, thermal-oxidative degradation, and hydrolysis. Even though several reaction pathways are already established, the understanding of specific degradation mechanisms for a given polymer system are still lacking. In this context, it is relevant to further combine experimental techniques commonly used to characterize polymer degradation as well as to further combine these techniques with modeling. Special attention should be paid to chromatography and rheometry, as they can be seen as sensitive techniques to capture molecular changes, even at initial stages of degradation. 

Case studies on the degradation of polyesters, styrene-based polymers, and polyolefins using conventional processing have been addressed. They aimed at assessing the effects of processing parameters on the degradation during processing. Among these parameters, the most important are the processing temperature, processing speed, feed rate, screw type, and screw elements. Molecular properties have been also explored, especially the initial average molar mass of the polymer, and in some situations, the grade of material. Other important insights have been provided by recyclability studies that assess the effects of multiple extrusion on the properties of the polymer materials. Overall, it can be stated that parameters which increase the residence time will induce a higher degree of degradation of the material. Furthermore, the processing temperature must be carefully chosen, as at too elevated temperatures the polymer materials are becoming too prone to degradation. 

The volume of literature concerning polymer degradation during extrusion-based additive manufacturing (EAM) is remarkably smaller than during conventional processing, which can be partially understood by their more recent development. However, research covering degradation during conventional processing in the past decades may also be incorporated, at least to certain extent, for these newer manufacturing techniques. It is of note that FFF includes two processing steps, which may further contribute to the degradation of the printed polymeric materials, as they are subjected to factors which induce degradation twice prior to their use.

## Figures and Tables

**Figure 1 molecules-28-02344-f001:**
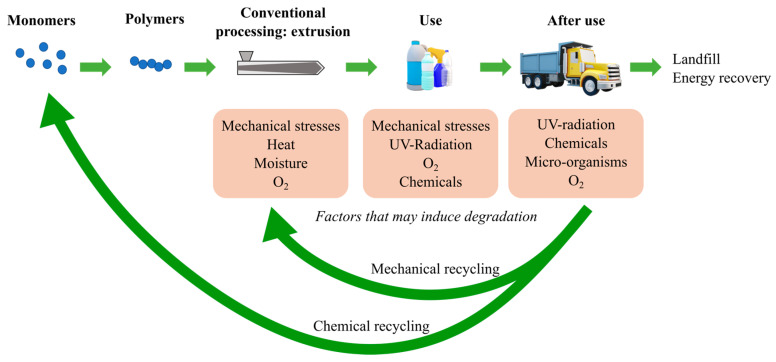
Summary of the most common factors which cause degradation of polymers during the life cycle of polymer goods, also including mechanical recycling in view of circularity. Chemical recycling is added for completeness and in that case, it is a desired degradation. In any case a detailed understanding of the relevance of molecular changes, that being the focus of the current contribution, is needed.

**Figure 2 molecules-28-02344-f002:**
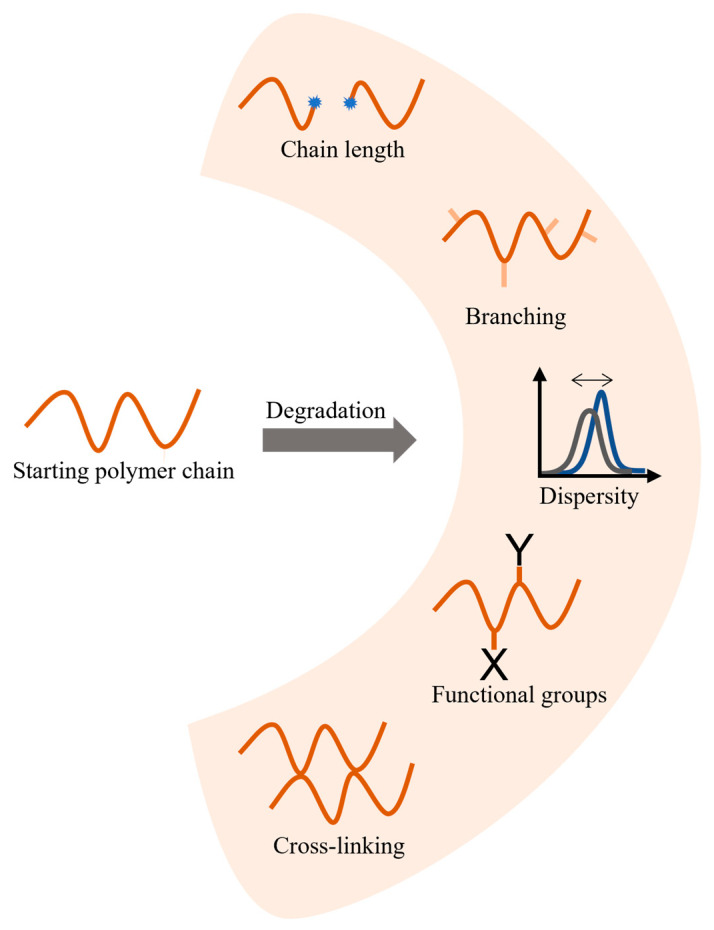
Typical molecular changes due to polymer degradation. Stabilizers can be added to avoid or minimize certain reactions.

**Figure 3 molecules-28-02344-f003:**
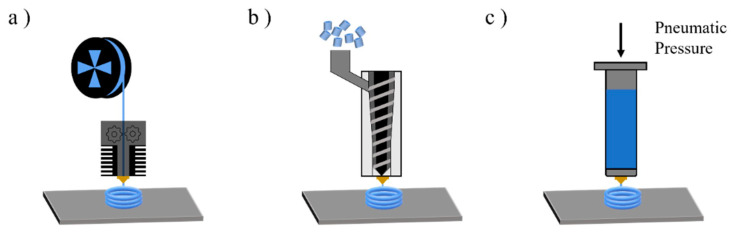
Methods for extrusion-based additive manufacturing (EAM) with (**a**) fused filament fabrication, (**b**) pellet-based additive manufacturing, and (**c**) syringe-based additive manufacturing.

**Figure 4 molecules-28-02344-f004:**
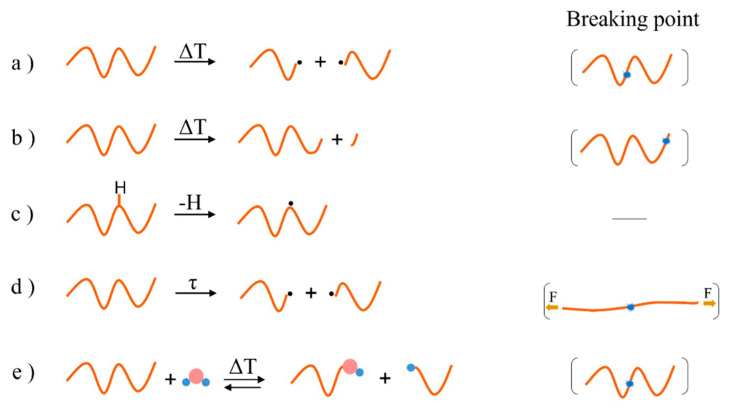
Examples of major pathways in polymer degradation, according to a variation in temperature (T) and mechanical stress (τ) or due the presence of water. (**a**) Thermal chain fission (also denoted as scission), (**b**) end-chain β-scission, (**c**) hydrogen abstraction, (**d**) mechanical-stress-induced chain fission, and (**e**) hydrolysis.

**Figure 5 molecules-28-02344-f005:**
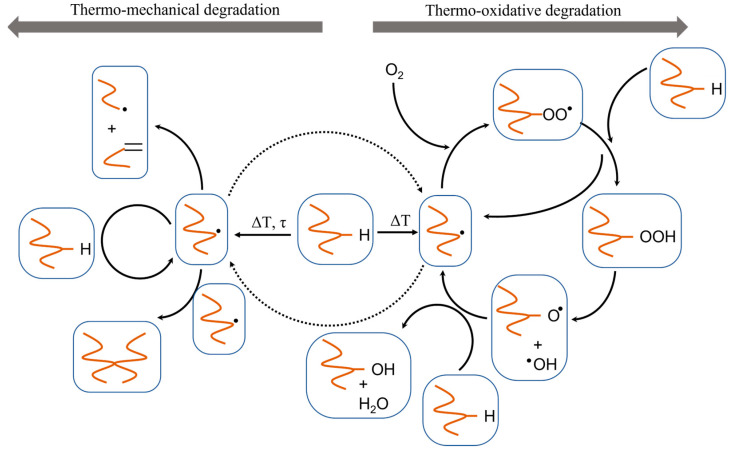
General scheme of thermo-mechanical and thermal-oxidative degradation of polymers, as based on Edeleva et al. [58] starting for illustration purposes from mid-chain radical formation.

**Figure 6 molecules-28-02344-f006:**
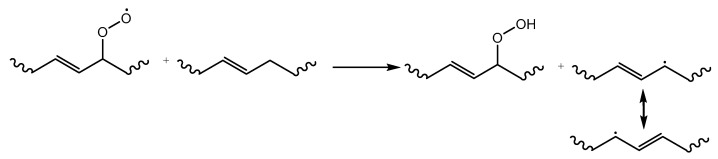
Hydrogen abstraction by a peroxyl radical from an unsaturated polymer chain, forming a hydroperoxide and a stable polymeric radical, as based on [63].

**Figure 7 molecules-28-02344-f007:**
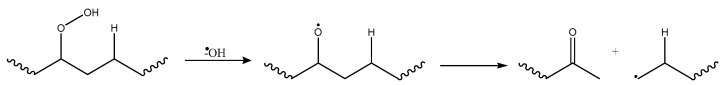
Example of β-scission reaction, starting by the presence of hydroperoxide molecules as formed in the presence of oxygen, as based on [64].

**Figure 8 molecules-28-02344-f008:**
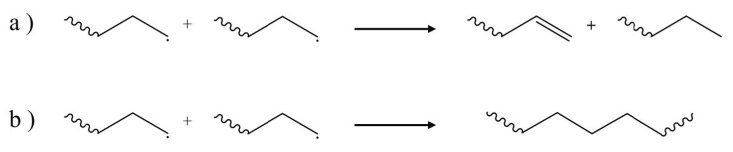
Example of most important termination pathways with less oxygen present: (**a**) disproportionation and (**b**) combination.

**Figure 9 molecules-28-02344-f009:**
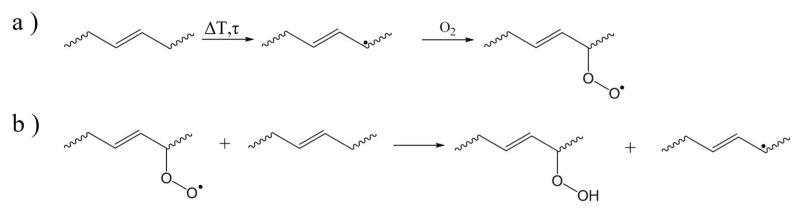
Key steps in thermal-oxidative degradation mechanism of ABS: (**a**) degradation initiation, (**b**) hydrogen abstraction leading to the formation of hydroperoxides, (**c**) possible products following the hydroperoxide decomposition, and (**d**) termination considering the formation of crosslinked structures.

**Figure 10 molecules-28-02344-f010:**
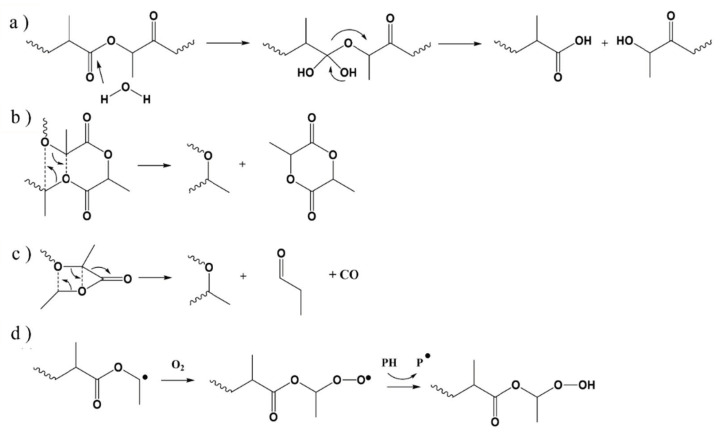
(**a**) Hydrolysis of PLA [111]. (**b**,**c**) Nonradical reactions of PLA degradation [116]. (**d**) Hydroperoxide formation and decomposition during the thermal oxidation of PLA [118].

**Figure 11 molecules-28-02344-f011:**
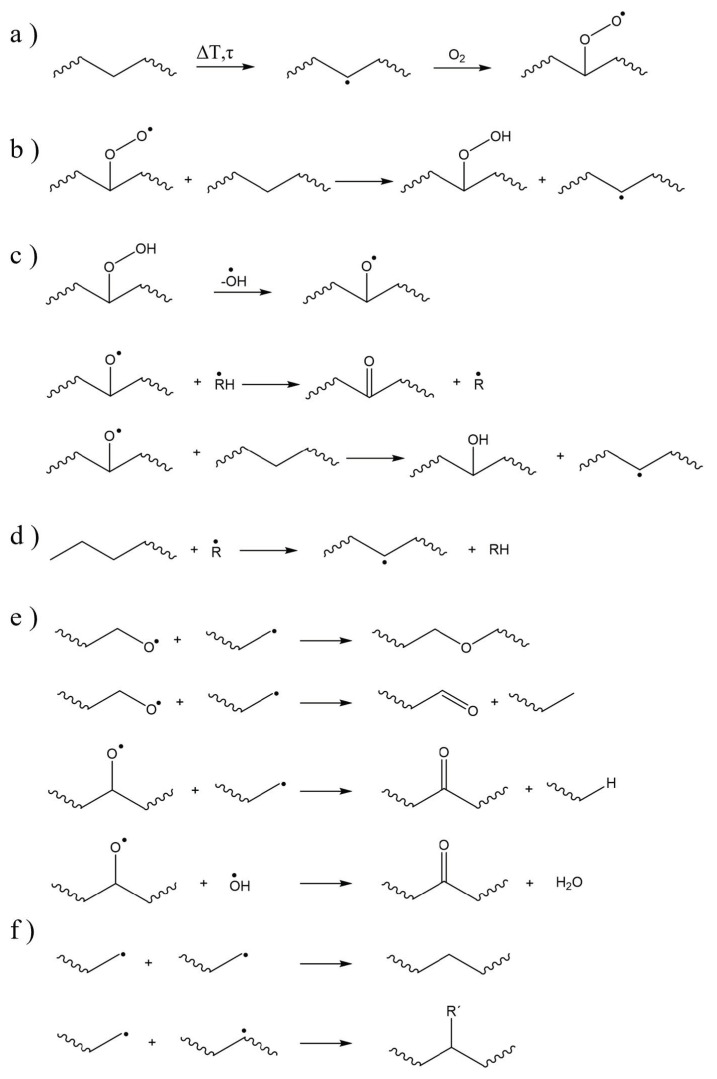
Examples of thermal-oxidative degradation reactions for polyethylene including (**a**) degradation initiation via a general chain transfer, (**b**) hydrogen abstraction leading to the formation of hydroperoxides, (**c**) possible products following the hydroperoxide decomposition, (**d**) auto-oxidation of PE, (**e**) possible termination with alkoxy radicals, and (**f**) termination of alkyl radicals [122]. R• being an alkyl radical, RO•, ROO•, or HO•.

**Figure 12 molecules-28-02344-f012:**
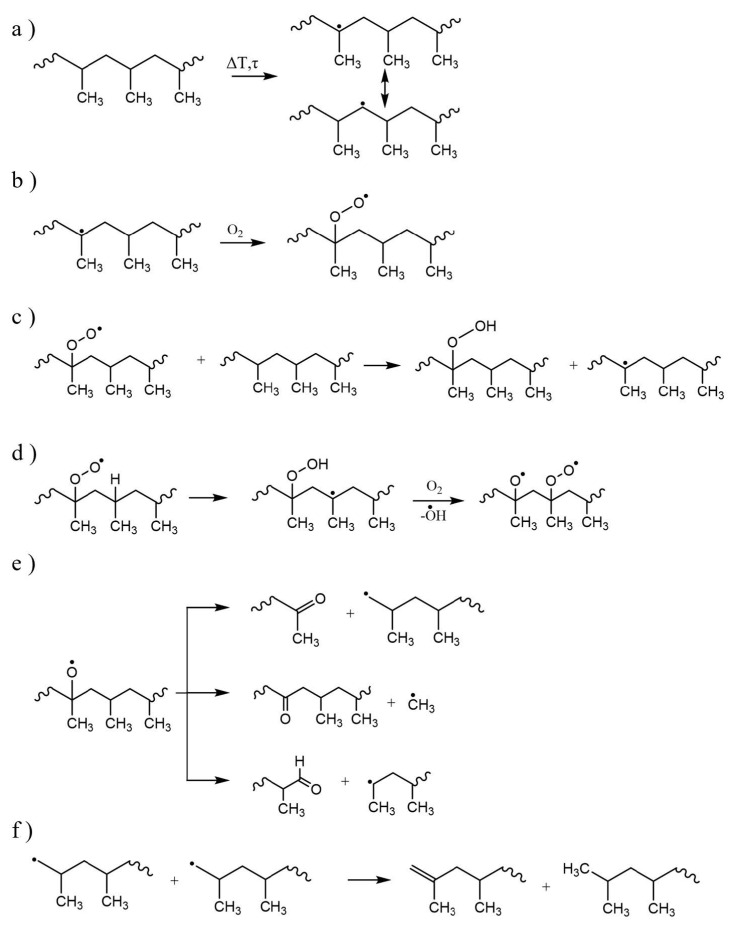
Examples of thermal-oxidative degradation reactions for polypropylene. (**a**) Degradation initiation, (**b**) reaction of radical with oxygen, (**c**) intermolecular and (**d**) intramolecular hydroperoxide formation, (**e**) formation of oxygenated functional groups, and (**f**) termination.

**Figure 13 molecules-28-02344-f013:**
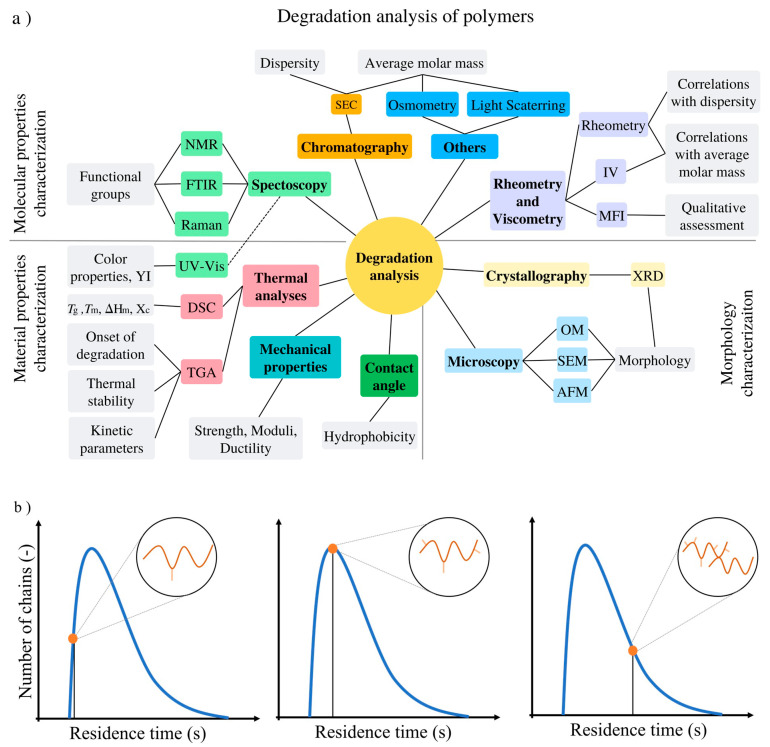
(**a**) Experimental characterization techniques to assess polymer degradation during processing, either on the molecular, morphological, or material level; SEC: size exclusion chromatography; NMR: nuclear magnetic resonance; FTIR: Fourier-transform infrared; IV: intrinsic viscosity; MFI: melt flow index; UV-Vis: ultraviolet-visible spectrophotometry; DSC: differential scanning calorimetry; TGA: thermogravimetric analysis; YI: Yellowness index; *T_g_*: glass transition temperature; *T_m_*: melt temperature; *T_c_*: crystallization temperature; Δ*H_m_*: enthalpy of fusion; *X_c_*: degree of crystallinity; XRD: X-ray diffraction; OM: optical microscopy; SEM: scanning electron microscopy; and AFM: atomic force microscopy. (**b**) Complementary strength of modeling tools focusing on coupled matrix-based Monte Carlo simulations [125,126,138], delivering the molecular structure of individual molecules (here only one shown) according to their residence time during extrusion-based mechanical recycling.

**Figure 14 molecules-28-02344-f014:**
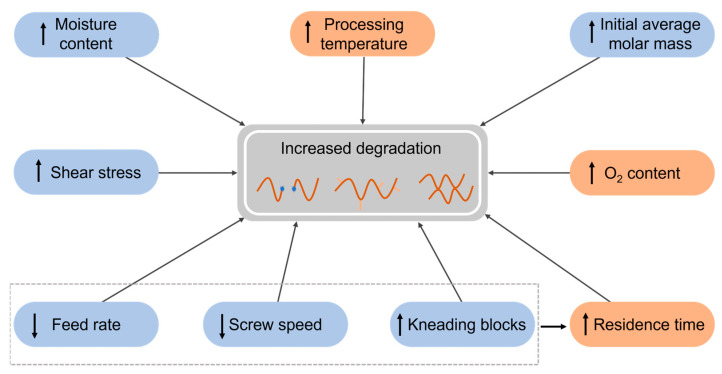
Summary of the influence of processing parameters on the degradation of polymers, with the most important parameters highlighted in orange.

**Table 1 molecules-28-02344-t001:** Most important degradation mechanisms during the life cycle of polymer goods.

Main Factor (s)	Degradation Mechanism
Temperature	Thermal degradation
Temperature and mechanical stresses	Thermo-mechanical degradation
Temperature and oxygen	Thermal-oxidative degradation
Water and temperature	Hydrolysis
UV radiation and oxygen	Photo-oxidative degradation
Chemicals	Chemical degradation
Micro-organisms	Biodegradation

**Table 2 molecules-28-02344-t002:** Bond dissociation energy values in kJ mol^−1^ relevant for many polymers [43].

Bond	Aromatic or Heterocyclic	Aliphatic
C-C	410	284–368
C=C	-	615
C-H	427–435	381–410
C-Cl	-	326
C-F	-	452
C-O	448	350–389
C-N	460	293–343
C=N	-	615

**Table 3 molecules-28-02344-t003:** Main degradation types of common polymers used for extrusion-based additive manufacturing (EAM); chain length must be read as average chain length.

Group	Name and Main Chemical Structure	Degradation Mechanism	Main Outcome of Degradation
Styrene-based materials(Rubber-modified)	Acrylonitrile butadiene styrene (ABS)	Thermal oxidation of butadiene units andThermo-mechanical	-Decrease of chain length due to chain fission [88]-Increase of chain length due to crosslinking [88,89]
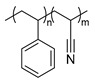	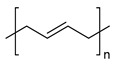
Styrene-Acrylonitrile (SAN)	(*trans*-1,4) Polybutadiene (PB)
High-impact polystyrene (HIPS)
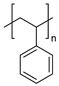	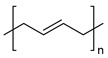
Polystyrene (PS)	(*trans*-1,4) Polybutadiene (PB)
Polyesters	Polylactic acid (PLA)	Hydrolysis of ester groups andThermo-mechanical	-Decrease of chain length due to hydrolysis or due to chain fission [90,91]
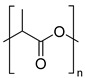
Polyethylene terephthalate glycol (PETG)
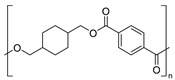
Polyolefins	Polypropylene (PP)	Thermal oxidation andThermo-mechanical	-Decrease of chain length due to chain fission [92,93].
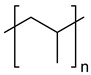
High-density polyethylene (linear)Low-density polyethylene (branched)	-Decrease of chain length due to chain fission [94,95]-Increase of chain length due to branching and crosslinking [94,95]
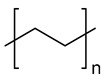
Polyaryletherketones	Polyether ether ketone (PEEK)	Thermal oxidation and Thermo-mechanical	-Increase of chain length due to branching and crosslinking [96,97]
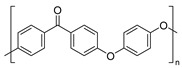
Polyamides	Polyamide 12	Thermal oxidation of C-H bonds adjacent to N-H groups andThermo-mechanical	-Decrease of chain length due to chain fission [98]-Increase of chain length due to post-polycondenstion and crosslinking [99]
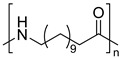
Polyamide 6,6
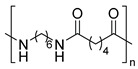

## Data Availability

Data are available upon reasonable request to the corresponding author.

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
