# Peer review of "Molecular Pathways for Polymer Degradation during Conventional Processing, Additive Manufacturing, and Mechanical Recycling"

_molecules, 2023, doi:10.3390/molecules28052344_

Round 1
Reviewer 1 Report
The authors have surveyed a range of literature, and reviewed the possible reaction mechanisms for degradation of various polymers during processing including AM. This topic is important for industrial application of plastics, as well as fundamental polymer science. The reviewer suggests this manuscript is worthy of publishing in MDPI Molecules. However, the reviewer would like to point out several issues to be fixed as follows.
Several technical terms seem to be inappropriate in polymer science field.
- molar mass -> molecular weight
- M_m -> M_w (weight-averaged molecular weight)
- dispersity -> polydispersity
- fission -> scission
(P.2, Fig. 1) This figure seems to be not suitable for this manuscript, because the present manuscript is focused on the degradation during the processing. Consider to revise it, or add comments in the main text on the entire material life cycle of plastics.
(P.8, Table 3) The contents in "Degradation mechanism" and "Main outcome of degradation" columns seem to be somewhat overlapped and redundant. Consider to merge the overlapped phrases to highlight the difference between these polymers.
(P.9, Fig. 9) Please reorganize this reaction scheme.
(a) The second line should follow the first line.
(c and d) The reactant is the same. Consider to merge, and show by radial arrows.
In addition, the chemical structure and the fonts seem to be small, and the dots for the radicals are hard to see.
(P.10, L.261) There remains an error.
(P.11, Figure 11) Consider to revise it as in Fig.9.
(P.14, L.403) "spectroscopy" should be inserted after (FTIR).
(P.20, Fig. 14) As mentioned in Conclusion, the residence time and the O2 content are the major factors. Consider to highlight importance of these dominant parameters.
(P.21, L.750-759) The first paragraph of Conclusion seems to be just a repetition.
Finally, the reviewer suggests that this manuscript should be checked by natives.
Author Response
Comments reviewer 1:
The authors have surveyed a range of literature, and reviewed the possible reaction mechanisms for degradation of various polymers during processing including AM. This topic is important for industrial application of plastics, as well as fundamental polymer science. The reviewer suggests this manuscript is worthy of publishing in MDPI Molecules. However, the reviewer would like to point out several issues to be fixed as follows.
Answer to general comment
We thank the reviewer for the general appreciation. We have accounted for all specific comments.
- Several technical terms seem to be inappropriate in polymer science field.
- molar mass -> molecular weight
- M_m -> M_w (weight-averaged molecular weight)
- dispersity -> polydispersity
- fission -> scission
Answer: We are somewhat puzzled by the comment of this reviewer. Based on IUPAC “Compendium of Polymer Terminology and Nomenclature” [1], the term Molar Mass should be used. “1.2 molar mass, M Mass divided by amount of substance. Note 1: Molar mass is usually expressed in g mol-1 or kg mol-1 units”. Whereas the molecular weight “is a pure number and must not be associated with any units”. In addition, when it comes to Polydispersity, IUPAC has replaced this term by Dispersity as discussed in [2].
- Compendium of Polymer Terminology and Nomenclature; Jones, R.G., Wilks, E.S., Metanomski, W.V., Kahovec, J., Hess, M., Stepto, R., Kitayama, T., Eds.; Royal Society of Chemistry: Cambridge, 2009; Vol. 0; ISBN 978-0-85404-491-7.
- Gilbert, R.G.; Hess, M.; Jenkins, A.D.; Jones, R.G.; Kratochvíl, P.; Stepto, R.F.T. Dispersity in polymer science (IUPAC recommendations 2009). Pure Appl. Chem. 2009, 81, 351–353, doi:10.1351/PAC-REC-08-05-02.
We have although decided to mention both terms.
Regarding fission vs scission we can understand the reviewer but strictly the term fission is the desired term as β-scission is often generalized as scission which it is not (radical start or not). We have made this clear in the revised manuscript.
- (P.2, Fig. 1) This figure seems to be not suitable for this manuscript, because the present manuscript is focused on the degradation during the processing. Consider to revise it, or add comments in the main text on the entire material life cycle of plastics.
Answer: The text was updated with comments on the life cycle of polymer goods. Also it has been made clearer that degradation reactions are key in the preferred routes of mechanical and chemical recycling.
- (P.8, Table 3) The contents in "Degradation mechanism" and "Main outcome of degradation" columns seem to be somewhat overlapped and redundant. Consider to merge the overlapped phrases to highlight the difference between these polymers.
Answer: It was considered to merge the overlapped phrases. However, after careful assessment, this would require grouping the polymers based on the outcome of degradation, instead of “polymer group” as not all of them behave in the same manner. In addition, LDPE and HDPE have different possible outcomes for degradation (on average), therefore merging the phrases could be misleading to the author. Based on this explanation, the authors have decided to keep the table as it is.
- (P.9, Fig. 9) Please reorganize this reaction scheme.
(a) The second line should follow the first line.
(c and d) The reactant is the same. Consider to merge, and show by radial arrows.
In addition, the chemical structure and the fonts seem to be small, and the dots for the radicals are hard to see.
Answer: The scheme was updated accordingly. We thank the reviewer for noting this.
- (P.10, L.261) There remains an error.
Answer: Text updated.
- (P.11, Figure 11) Consider to revise it as in Fig.9.
Answer: The scheme was updated accordingly.
- (P.14, L.403) "spectroscopy" should be inserted after (FTIR).
Answer: Updated.
- (P.20, Fig. 14) As mentioned in Conclusion, the residence time and the O2 content are the major factors. Consider to highlight importance of these dominant parameters.
Answer: The key parameters are now highlighted in the figure, including processing temperature.
- (P.21, L.750-759) The first paragraph of Conclusion seems to be just a repetition.
Answer: In the first paragraph the review work is summarized, and important inputs are given. Therefore, we believe it is important for the overall conclusion of the work.
- Finally, the reviewer suggests that this manuscript should be checked by natives.
Answer: Additional checks has been performed as specifically in the introduction part some sentences needed to be made more fluently.
Reviewer 2 Report
The authors made a very good effort to summarize the degradation mechanisms of several key polymer materials currently used in the industry to help with post use procession and better recycling towards a viable Circular Economy (CE) model. Overall, I believe that the manuscript contributes to a great extend in the field of plastic post consumption processing/recycling and CE. I think that a moderate grammar and spell revision should be made by the authors to improve readability. Also, adding a few more citation in certain areas would vastly help the reader.
· Grammar and Spelling Comments
Line 38; Please correct the sentence as follows: “In general, materials subjected to multiple-step processing techniques are more prone to degradation…”. The way it is currently written implies that the techniques are prone to degradation, which is not true, they induce degradation.
Line 66; Please consider splitting the sentence as follows: “…place at the molecular scale. Therefore, a good knowledge on polymer chemistry…”.
Line 68; Please replace “in e.g.” with “such as”.
In figure 2, I believe the graph presented as case 3 in the right part of the schematic, molecular changes upon degradation, should be labeled as “dispersity” and not “chain length”, since chain length effects are depicting in case 1.
Line 88; Please replace the “In other to do so” with “In order to do so”.
Line 115; Please consider the following changes: “…and we further discuss zoom in on the degradation shared mechanisms of selected polymers as widely used…”.
Line 118; Please correct the following: “In a the third part, case…” (replace “a” with “the”.
Line 140; Please remove the words “although” and “also” from the sentence.
Line 141; Add the word also as follows: “…for chain transfer also mater.”.
Line 142; Please correct “chain fission” to “chain scission”.
Line 144; Please correct the word “fission” to “scission”. Fission is a nuclear reaction process. I believe the focus here is backbone bond-scission effects that break the polymer main chains resulting in lower molecular weight products. Please do a document “find” search and replace the word fission with scission were appropriate.
Line 458; Please make the following changes to the sentence: “Thermal analysis is broadly applied…”.
· Content Related Comments
Lines 151-154; Side-group elimination, such as hydrogen abstraction, can also result in crosslinking between free radicals in adjacent polymer chains (termination reaction). Furthermore, two macroradicals on adjacent polymer chains produced from hydrogen or side-group elimination can terminate producing unsaturations through disproportionation reactions. Please consider adding those to the manuscript (in line 154) with citations. I also recommend adding more citations to this section (lines 144 – 154) to showcase how and in what conditions certain degradation reactions are favored over others, there is a vast amount of literature discussing those thermal decomposition reactions in polymers.
In figure 5, on the right part of the figure, where thermal-oxidative degradation is shown another possible reaction (which is a propagation reaction) is the decomposition of peroxyl radicals that can lead to chain scission. This reaction is important and will produce lower molecular weight chains, new macroradicals, and carbon dioxide. Please consider adding this reaction to the figure to enhance it by showing all the possible reactions, or simply add it to the manuscript as done for the reaction in figure 7, for example.
Line 261; Please find and add the correct reference for this statement.
In Figure 11 and the related text, the thermal decomposition of hydroperoxides (shown in figure 11c), besides the presented formation of alkoxy and hydroxyl radicals (which is a propagation reaction), can also result in the formation of ketones and water molecules (termination reaction). Please consider adding that to the termination reactions of this section.
Figure 13. Could the authors please improve the resolution of this figure. A lot of the text is hard to read.
Author Response
Comments reviewer 2:
The authors made a very good effort to summarize the degradation mechanisms of several key polymer materials currently used in the industry to help with post use procession and better recycling towards a viable Circular Economy (CE) model. Overall, I believe that the manuscript contributes to a great extend in the field of plastic post consumption processing/recycling and CE. I think that a moderate grammar and spell revision should be made by the authors to improve readability. Also, adding a few more citation in certain areas would vastly help the reader.
Answer to general comment
We thank the reviewer of the nice words. We have updated the text as requested.
- Line 38; Please correct the sentence as follows: “In general, materials subjected to multiple-step processing techniques are more prone to degradation…”. The way it is currently written implies that the techniques are prone to degradation, which is not true, they induce degradation.
Answer: We can understand the confusing. We have made a changes “In general, multiple-step processing techniques are more prone to induce polymer degradation, a more recent example being additive manufacturing (AM) or 3D printing via fused filament fabrication (FFF).”
- Line 66; Please consider splitting the sentence as follows: “…place at the molecular scale. Therefore, a good knowledge on polymer chemistry…”.
Answer: The change has been performed.
- Line 68; Please replace “in e.g.” with “such as”.
Answer: The change has been performed.
- In figure 2, I believe the graph presented as case 3 in the right part of the schematic, molecular changes upon degradation, should be labeled as “dispersity” and not “chain length”, since chain length effects are depicting in case 1.
Answer: Thanks for pointing it out. We changed it accordingly.
- Line 88; Please replace the “In other to do so” with “In order to do so”.
Answer: The change has been performed.
- Line 115; Please consider the following changes: “…and we further discuss zoom in on the degradation shared mechanisms of selected polymers as widely used…”.
Answer: The sentence was changed to “We initially focus on the common degradation pathways involved in polymer processing. We further discuss the degradation mechanisms of selected polymers that are widely used in EAM.”
- Line 118; Please correct the following: “In a the third part, case…” (replace “a” with “the”.
Answer: The change has been performed.
- Line 140; Please remove the words “although” and “also” from the sentence.
Answer: The change has been performed..
- Line 141; Add the word also as follows: “…for chain transfer also mater.”.
Answer: The change has been performed.
- Line 142; Please correct “chain fission” to “chain scission”.
Answer: See next comment
- Line 144; Please correct the word “fission” to “scission”. Fission is a nuclear reaction process. I believe the focus here is backbone bond-scission effects that break the polymer main chains resulting in lower molecular weight products. Please do a document “find” search and replace the word fission with scission were appropriate.
Answer: We have highlighted the double naming in the text but would like to state that scission is often seen as equivalent or related to β-scission and these are chemically different. One has a neutral molecule at the start and that is why in the field fission has been used as well (as in the nuclear reactions). A relevant reference is: Ind. Eng. Chem. Res. 2003, 42, 2722-2735
- Line 458; Please make the following changes to the sentence: “Thermal analysis is broadly applied…”.
Answer: Ok.
Content Related Comments
- Lines 151-154; Side-group elimination, such as hydrogen abstraction, can also result in crosslinking between free radicals in adjacent polymer chains (termination reaction). Furthermore, two macroradicals on adjacent polymer chains produced from hydrogen or side-group elimination can terminate producing unsaturations through disproportionation reactions. Please consider adding those to the manuscript (in line 154) with citations. I also recommend adding more citations to this section (lines 144 – 154) to showcase how and in what conditions certain degradation reactions are favored over others, there is a vast amount of literature discussing those thermal decomposition reactions in polymers.
Answer: The text has been updated accordingly.
- In figure 5, on the right part of the figure, where thermal-oxidative degradation is shown another possible reaction (which is a propagation reaction) is the decomposition of peroxyl radicals that can lead to chain scission. This reaction is important and will produce lower molecular weight chains, new macroradicals, and carbon dioxide. Please consider adding this reaction to the figure to enhance it by showing all the possible reactions, or simply add it to the manuscript as done for the reaction in figure 7, for example.
Answer: We understand the comment of the reviewer. We have made the link with Figure 7 in the caption.
- Line 261; Please find and add the correct reference for this statement.
Answer: Text updated.
- In Figure 11 and the related text, the thermal decomposition of hydroperoxides (shown in figure 11c), besides the presented formation of alkoxy and hydroxyl radicals (which is a propagation reaction), can also result in the formation of ketones and water molecules (termination reaction). Please consider adding that to the termination reactions of this section.
Answer: The scheme has been updated accordingly.
- Figure 13. Could the authors please improve the resolution of this figure. A lot of the text is hard to read.
Answer: The figure has been updated.
Reviewer 3 Report
This comprehensive study provides a broad overview of polymer degradation mechanisms during conventional (e.g., extrusion and injection molding) and emerging (e.g., additive manufacturing; AM) techniques. In addition, the authors provide a summary of the main characterization techniques essential for stated investigations, while case studies regarding different types of polymers are also given. The literature is up to date and thoroughly investigated, with a share of self-citations of some 10%. Some minor typographical (such as double spaces) in the manuscript should be corrected, while some language polishing might be beneficial. Finally, I find it necessary for the authors to use the term chain scission instead chain fission throughout the manuscript, as it is common in the field of polymer science.
In my professional opinion, no further changes are necessary, and I recommend this manuscript for publication after making a few indicated minor corrections.
Author Response
Comments reviewer 3:
This comprehensive study provides a broad overview of polymer degradation mechanisms during conventional (e.g., extrusion and injection molding) and emerging (e.g., additive manufacturing; AM) techniques. In addition, the authors provide a summary of the main characterization techniques essential for stated investigations, while case studies regarding different types of polymers are also given. The literature is up to date and thoroughly investigated, with a share of self-citations of some 10%.
Answer:
We thank the reviewer for the general appreciation.
- Some minor typographical (such as double spaces) in the manuscript should be corrected, while some language polishing might be beneficial.
Answer: The text has been revised for such typographical errors
- Finally, I find it necessary for the authors to use the term chain scission instead chain fission throughout the manuscript, as it is common in the field of polymer science.
Answer: We have highlighted the double naming in the text but would like to state that scission is often seen as equivalent or related to β-scission and these are chemically different. One has a neutral molecule at the start and that is why in the field fission has been used as well (as in the nuclear reactions). A relevant reference is: Ind. Eng. Chem. Res. 2003, 42, 2722-2735.
- In my professional opinion, no further changes are necessary, and I recommend this manuscript for publication after making a few indicated minor corrections.